# XLM-V: Overcoming the Vocabulary Bottleneck in Multilingual Masked Language Models

**Davis Liang**     **Hila Gonen**     **Yuning Mao**     **Rui Hou**

**Naman Goyal**   **Marjan Ghazvininejad**   **Luke Zettlemoyer**   **Madian Khabsa**

Meta AI

## Abstract

Large multilingual language models typically rely on a single vocabulary shared across 100+ languages. As these models have increased in parameter count and depth, vocabulary size has remained largely unchanged. This *vocabulary bottleneck* limits the representational capabilities of multilingual models like XLM-R. In this paper, we introduce a new approach for scaling to very large multilingual vocabularies by de-emphasizing token sharing between languages with little lexical overlap and assigning vocabulary capacity to achieve sufficient coverage for each individual language. Tokenizations using our vocabulary are typically more semantically meaningful and shorter compared to XLM-R. Leveraging this improved vocabulary, we train XLM-V, a multilingual language model with a one million token vocabulary. XLM-V outperforms XLM-R on every task we tested on ranging from natural language inference (XNLI), question answering (MLQA, XQuAD, TyDiQA), to named entity recognition (WikiAnn). XLM-V is particularly effective on low-resource language tasks and outperforms XLM-R by 11.2% and 5.8% absolute on MasakhaNER and Americas NLI, respectively.

## 1 Introduction

While multilingual language models have increased in parameter count and depth over time, vocabulary size has largely remained unchanged: mBART (680M parameters; Liu et al. 2020), XGLM (7.5B parameters, Lin et al. 2021), XLM-R XXL (10.7B parameters; Goyal et al. 2021), mT5 XXL (13B parameters; Xue et al. 2020); and BLOOM (176B parameters; Scao et al. 2022) all share the same 250K token vocabulary size as XLM-R base (Conneau et al., 2019), a 250M parameter model.

For models like mT5 and XLM-R, this 250K vocabulary is shared across 100+ languages. Discounting shared tokens, this results in an average of

2,500 unique tokens per language, calling into question the vocabulary's ability to represent the diverse selection of languages that it was intended to model. For example, there are 8,105 characters in the Table of General Standard Chinese characters and over 100,000 unique characters in total; the number of commonly used Chinese words (consisting of multiple characters) is even larger (Wikipedia, 2023). In fact, prior work has already shown that this vocabulary bottleneck hinders the performance of multilingual models on question answering and sequence labeling where in-depth token-level and sequence-level understanding is essential (Wang et al., 2019).

In this paper, we construct a large multilingual vocabulary by attending to two core principles: (1) vocabularies can be improved by de-emphasizing token sharing between languages with little lexical overlap and (2) proper vocabulary capacity allocation for individual languages is crucial for ensuring that diverse languages are well-represented. Then, we show that our new vocabulary exhibits favorable characteristics including the ability to frequently output semantically meaningful tokenizations while reducing over-tokenization for low-resource languages. Finally, we present XLM-V, the first multilingual language model with a one million token vocabulary trained on 2.5TB of data from Common Crawl (Conneau et al., 2019).

Our main contributions are as follows:

- In Section 3, we present our method for constructing large multilingual vocabularies. Specifically, we improve upon the language clustering algorithm from Chung et al. (2020) by constructing better vector representations for individual languages and leverage Zheng et al. (2021) to improve the vocabulary capacity assignments for each cluster.

- In Section 5, we demonstrate that XLM-V outperforms comparable baselines that have

the same vocabulary size on XNLI. Additionally, XLM-V outperforms XLM-R on every multilingual language understanding task we tested on (including XNLI, WikiAnn, MLQA, XQuAD, and TyDiQA) by an average of 3.5 points absolute. XLM-V performs especially well on low-resource evaluation datasets like AmericasNLI and MasakhaNER, outperforming XLM-R by 5.8% absolute accuracy and 11.2% absolute F1, respectively.

- Finally, in Section 6, we provide examples and quantitative analysis to compare our new vocabulary to various baselines. Most notably, we provide evidence showing that expanding the vocabulary beyond 1M tokens can *degrade* performance on downstream tasks.

## 2   Background

### 2.1   Sentencepiece

The Unigram Language Model (ULM) from Kudo and Richardson (2018) is a popular subword segmentation algorithm used to construct vocabularies. ULM begins with a large initial vocabulary that is iteratively pruned to maximize the likelihood of the training corpus (under a unigram language model of the tokens) until the number of tokens falls below some pre-determined vocabulary size threshold, $|V|$. During tokenization, ULM decodes the most probable segmentation of a sequence through the Viterbi algorithm (Viterbi, 1967). This method is used by both XLM-R and our work.

### 2.2   Clustering

Chung et al. (2020) proposed an approach to multilingual vocabulary construction that balances the trade-off between optimizing for cross-lingual subword sharing and the need for robust representation of individual languages.

Their procedure for building a multilingual vocabulary contains several steps. First, the authors train individual sentencepiece models for each language: for each language $l$ in the set of languages $L$, a vocabulary $V^l$ is generated. Then, they create the shared lexicon $V^L$ by taking the union of each language-specific vocabulary, $V^L = \cup_{l \in L} V^l$. Next, for each language $l$, they construct a binary vector $v^l$ of dimension $|V^L|$ which represents the lexicon of $l$. Each component of $v^l$ corresponds to a subword in $V^L$. In other words, the binary vector $v^l$ contains a 1 corresponding to each subword

present in the vocabulary of $l$. An illustration of this step is shown in Figure 1. Then, the authors cluster the binary vectors to group lexically similar languages together. Finally, they construct a vocabulary for each cluster and combine the per-cluster vocabularies together to form a unified multilingual vocabulary.

### 2.3   Vocabulary allocation

Zheng et al. (2021) proposed the average log probability (ALP) to evaluate the ability of a vocabulary to represent a particular language. Specifically, given a monolingual corpus composed of sentences $\mathcal{D}_i = \{s_1, ..., s_{|\mathcal{D}_i|}\}$ from the $i$-th language and tokenized with vocabulary $V$, the average log probability is defined as;

$$ALP(\mathcal{D}_i, V) = \frac{1}{|\mathcal{D}_i|} \sum_{j=1}^{|\mathcal{D}_i|} \sum_{k=1}^{|s_j|} \log p_{uni}(s_j^k) \quad (1)$$

where $s_j^k$ is the $k$-th subword of the sentence $s_j$ and $p_{uni}(\cdot)$ is the unigram distribution counted on the monolingual corpus $\mathcal{D}_i$. The authors first show that ALP is highly correlated with downstream task performance and then propose a greedy algorithm to determine the desired vocabulary capacity for individual languages in the multilingual vocabulary.

## 3   Methodology

### 3.1   Building the vocabulary

In this subsection, we describe our method for constructing multilingual vocabularies. At a high level, we (1) train individual monolingual sentencepiece models (SPM) for each language in our dataset using the Unigram Language Model (ULM) algorithm (Kudo and Richardson, 2018), (2) use the per-language vocabularies to construct lexical representation vectors for each language, (3) cluster the lexical representation vectors using K-Means, assign vocabulary capacities for each cluster using the ALP, and then construct per-cluster vocabularies using the ULM algorithm, and (4) create the final multilingual vocabulary by taking the union of the vocabularies for each cluster.

**Training monolingual SPMs**   To acquire the data for building the vocabulary, we perform sampling with temperature $t = 2$ to sample 1 billion lines of text from CC100 (up-sampling lower-resource and down-sampling data from high resource languages). Then, for each language in CC100, we

| Cluster | $|V^c|$ | Languages |
|---|---|---|
| $c_1$ | 174,504 | fa, pa, sa, ka, ur, lo, my, ne, am, te, my, th, ta, ko, bn, ml, he, sd, as, hi, km, gu, kn, si, yi, mr, ps, or, xh, ar, ug |
| $c_2$ | 102,722 | ja, zh-TW, zh-CN |
| $c_3$ | 186,881 | fi, sk, om, sw, ln, az, lg, uz, so, hy, ss, hu, la, ff, et, ta, wo, lv, ku, te, sc, el, pl, lt, tr |
| $c_4$ | 110,148 | pt, eu, gl, gn, it, ca, qu, es |
| $c_5$ | 24,752 | af, li, nl, fy |
| $c_6$ | 19,801 | hr, sl, bs |
| $c_7$ | 101,485 | bg, ky, uk, be, kk, sr, mk, ru, mn |
| $c_8$ | 279,702 | su, jv, tl, sv, tn, no, id, ig, bn, ns, mg, cs, ms, ro, ur, rm, ha, ga, ht, is, eo, gd, br, hi, en, cy, fr, vi, da, yo, de, sq |

Table 1: Lexical clustering results for XLM-V with number of clusters $k = 8$ and a total vocabulary capacity of 1M.

train a language-specific sentencepiece model with a vocabulary size of 30,000 (per language) using this data.

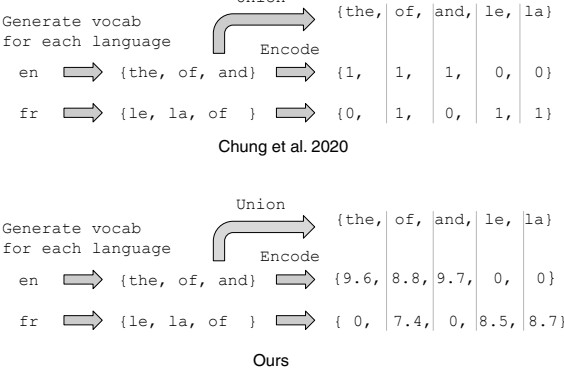

Figure 1: Similar to Chung et al. (2020), we also leverage the per-language sentencepiece vocabularies as a "lexical fingerprint" for clustering. However, instead of using binary vectors, we use the unigram log probability instead.

**Constructing lexical fingerprints** We then construct a vector representation of each language using the vocabularies of each language as shown in Figure 1. Unlike Chung et al. (2020), where a language is represented by a binary vector containing a 1 corresponding to each subword present in the vocabulary of that language, we instead use the negative log probability that each token appears in the respective language's monolingual corpus. We hypothesize that weighting each token by its likelihood of occurring better represents the lexical fingerprint of a language.

**Clustering and capacity allocation** Next, we construct language clusters and train sentencepiece models for each cluster in order to discourage the vocabulary sharing between lexically dissimilar languages. Before training per-cluster sentencepiece models, we need to first decide on the vocabulary size, or vocabulary capacity, to allocate to each cluster. Unfortunately, we found that the

method for assigning vocabulary capacities used by Chung et al. (2020) (i.e. proportionally to the set union of the per-language vocabularies in each cluster) resulted in several clusters with deficient vocabulary capacity. For example, cluster $c_2$ in Table 1 (a smaller cluster that contains lexically diverse languages: Chinese Simplified, Chinese Traditional, and Japanese), was assigned a capacity of just 28,593 tokens.

We instead use the per-language vocabulary capacity allocations (Zheng et al., 2021) optimized for the CC100 dataset. By doing so, the vocabulary capacity assigned to $c_2$ was increased to 102,722. For each tail-end (low-resource) language that was not covered in Zheng et al. (2021), we allocate a 2,000 token vocabulary budget. Rather than use the vocabulary allocations directly, we take their relative values and rescale them to sum up to the vocabulary capacity of our choosing (e.g. 1M, 2M, etc.). Finally, we perform K-Means clustering with $k = 8$, based on experiments from Chung et al. (2020) showing that $k = 8$ results in the best performance on downstream tasks. We expect the ideal number of clusters to vary not based on the number of languages but rather on the identity of those languages and their respective similarities to one another.

**The final vocabulary** For each resulting cluster, we train per-cluster sentencepiece models and combine the vocabularies of each cluster into a single multilingual vocabulary. The final vocabulary consists of 901,629 tokens (remaining 98,371 tokens overlapped between the 8 clusters), meaning that on average over 90% of the tokens learned in each cluster are unique.

### 3.2 Training the model

To pretrain our model, we follow the same training procedure from XLM-R (Conneau et al., 2019). Specifically, we use the CC100 dataset with a sampling temperature of 0.3 to increase the amount of low- and medium-resource language examples

seen during training. We use the Adam optimizer (Kingma and Ba, 2014) with the default $(\beta_1, \beta_2)$ and $\epsilon$ parameters of (0.9, 0.98) and 1e-6, respectively. We use a learning rate of 6e-4, a warmup of 15,000 steps, a batch size of 8,192 distributed across 256 A100 GPUs, and train for a total of 1.5M iterations. Each batch consists of examples concatenated up to the maximum sequence length of 512. We pretrain the model using the Masked Language Model (MLM) task (Devlin et al., 2018) with the standard masking rate of 15%.

Increasing the vocabulary size can significantly increase pretraining time due to the computationally intensive softmax layer. To address this, prior works have leveraged approximation tricks such as adaptive softmax (Baevski and Auli, 2018) and adaptive inputs (Joulin et al., 2017). However, we found that these tricks require non-trivial amounts of tuning and resulted in slower convergence and increased training instability. In this paper, we perform pretraining without any approximation tricks noting that this method may not be feasible when the vocabulary is scaled beyond 2M.[1]

## 4 Experiment setup

### 4.1 Baselines

Aside from training XLM-V, we also construct several baselines to compare our model against. To construct our baselines, we first create the respective vocabularies and then pretrain transformer encoders (12-layers, equivalent to XLM-R base) using these vocabularies. For the rest of the paper, we will use the following names to refer to the vocabulary and the model interchangeably.

**XLM-R (250K)** The XLM-R vocabulary is created using the same procedure from (Conneau et al., 2019) by applying the ULM algorithm described in Section 2 on a corpus of 1B lines of text sampled from CC100. The result is a multilingual vocabulary with 250,002 tokens. For our experiments, we simply re-use the publicly available XLM-R sentencepiece model and pretrained model checkpoint from fairseq (Ott et al., 2019).

**XLM-R (1M)** We construct a 1M token vocabulary by following the same approach as XLM-R (250K) with an increased vocabulary capacity.

**Chung et al. (2020) (1M)** We create a 1M token vocabulary using the lexical clustering approach from Chung et al. 2020 as described in Section 2.

### 4.2 Datasets

**CC100** (Conneau et al., 2019) is a multilingual corpus created from one Common Crawl dump for English and twelve dumps for all other languages. The resulting corpus contains 2.5 TB of data split between 116 languages. We use this dataset exclusively for constructing vocabularies and pretraining our models.

**FLoRes-200** (Goyal et al., 2022) is an evaluation corpus consisting of 3,001 sentences extracted from 842 English Wikipedia articles and covering a variety of different topics and domains. These sentences have been translated into 200 languages by professional translators through a carefully controlled process.

**XNLI** (Conneau et al., 2018) asks whether a premise sentence entails, contradicts, or is neutral toward a hypothesis sentence. Crowd-sourced English data is translated to 10 other languages by professional human translators and used for evaluation, while the Multi-Genre Natural Language Inference Corpus (MultiNLI) (Williams et al., 2018) data is used for training.

**MLQA** (Lewis et al., 2019) [2] is a QA evaluation dataset created by mining target language sentences that are parallel to sentences in English from Wikipedia, crowd-sourcing annotations in English, and translating the question and aligning the answer spans in one of the 6 target languages. It consists of over 12K QA instances in English and 5K in each other language. The training set of MLQA is SQuAD v1.1 (Rajpurkar et al., 2016).

**XQuAD** (Artetxe et al., 2019) translates the dev set of SQuAD v1.1 into 10 other languages through professional translators. The resulting dataset is used for evaluation. The training set of XQuAD is SQuAD v1.1.

**TyDiQA-GoldP** (Clark et al., 2020) is a question answering (QA) dataset covering 11 typologically diverse languages with 200K QA pairs. Questions

---

[1]For a model with a vocabulary size of 1M, each iteration of MLM pretraining took 2.5 times longer than the same model with a 250K token vocabulary.

[2]For the question answering tasks, instead of validating the selected spans after retrieving the n-best answers, *we propose to only retrieve n-best answer spans that are valid* (e.g. span start and end indices are part of the passage context). This change improves QA performance for both the baseline models and XLM-V.

| Model | XNLI Acc. | NER Acc. | MLQA EM / F1 | TyDiQA EM / F1 | XQuAD EM / F1 | ANLI F1 | MNER F1 | Average |
|---|---|---|---|---|---|---|---|---|
| XLM | 69.1 | - | 32.6 / 48.5 | 29.1 / 43.6 | 44.3 / 59.8 | - | - | - |
| XLM-R | 76.2 | - | 46.3 / 63.7 | - / - | - / - | 38.5 | - | - |
| XLM-R *reimpl.* | 74.9 | 61.3 | 46.7 / 64.4 | 38.3 / 56.0 | 56.0 / 71.3 | 39.6 | 20.9 | 55.5 |
| XLM-V | **76.0** | **64.7** | **47.7 / 66.0** | **39.7 / 56.9** | **56.3 / 71.9** | **45.4** | **32.1** | **59.0** |

Table 2: Overall results across multiple multilingual datasets comparing our model against the XLM and XLM-R baselines. All results are based on crosslingual transfer after fine-tuning on English data. We computed the average result using the accuracy or F1 of each task. "*reimpl*" is our re-implementation of finetuning, used by both XLM-R and XLM-V. Please refer to the appendix for specific hyperparameters to reproduce each result. EM stands for exact match. ANLI refers to AmericasNLI and MNER refers to MasakhaNER.

in TyDiQA are written without seeing the answers leading to significantly less lexical overlap than XQuAD or MLQA. The languages of TyDiQA are selected to be diverse with regard to their typology. We use the gold passage version of the Typologically Diverse Question Answering dataset.

**NER** (Pan et al., 2017) consists of 48 languages and is based on the WikiAnn (PAN-X) dataset. Named entities were automatically annotated with LOC, PER, and ORG tags through knowledge base properties, crosslingual and anchor links, self-training, and data selection. Similar to (Hu et al., 2020), we use the balanced dev and test splits from Rahimi et al. (2019).

**Americas NLI** (Ebrahimi et al., 2021) is an extension of XNLI to 10 indigenous languages of the Americas constructed by translating a subset of XNLI using human translators. These languages contain interesting linguistic features such as a rich system of applicative suffixes (Asháninka), directional verbs (Bribri), and nominal incorporation (Wixarika). Presently, these languages are written, spoken, and used in an official capacity by tens of thousands to several million people in Central and Southern America. The training set of Americas NLI is MultiNLI.

**MasakhaNER** (Adelani et al., 2021) is the first large, publicly available, and high-quality dataset for named entity recognition (NER) in ten African languages including Amharic, Hausa, Igbo, and others. The languages covered in this dataset have varied scripts and range from 4M to 98M speakers in regions across East, West, Central, and Northwest Africa.

# 5 Results

## 5.1 Comparisons using partial training

We first perform a study to measure the impact of our new vocabulary on downstream performance. Specifically, we pretrain a 12-layer transformer encoder model using Masked Language Modeling on the CC100 corpus for each baseline as well as for our proposed method. Because pretraining is expensive, we limit the batch size to 2,048 and the number of total steps to 300,000 for these experiments. The results in Figure 2 show that our model outperforms all baselines on XNLI including XLM-R (1M) by 1.34% and Chung et al. (2020) by 1.11% absolute accuracy.

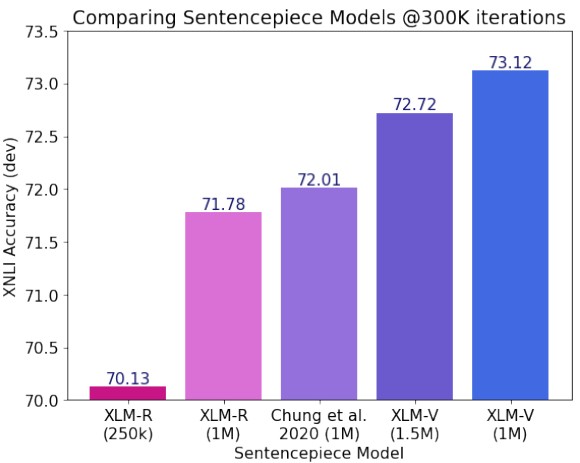

Figure 2: We compare the performance of the same model trained with different sentencepiece vocabularies. The models are all trained for 300K iterations with a batch size of 2,048 on the CC100 corpus.

## 5.2 Fully trained model

We evaluate an XLM-V (1M) model, trained on CC100 for 1.5M iterations with a batch size of 8,192, on several tasks including natural language inference (XNLI), question answering (MLQA,

| Model | en | fr | es | de | el | bg | ru | tr | ar | vi | th | zh | hi | sw | ur | AVG |
|---|---|---|---|---|---|---|---|---|---|---|---|---|---|---|---|---|
| *Finetune multilingual model on English training set (Cross-lingual Transfer)* | | | | | | | | | | | | | | | | |
| XLM-R *reimpl.* | 85.4 | 78.5 | 79.1 | 77.7 | 76.1 | 78.1 | 76.3 | 73.9 | 72.3 | 75.6 | **73.0** | 74.9 | 70.5 | 65.8 | 66.5 | 74.9 |
| XLM-V | **85.6** | **79.6** | **79.5** | **78.4** | **76.9** | **79.6** | **76.6** | **74.0** | **73.1** | **76.2** | **73.0** | **75.1** | **72.0** | **70.5** | **69.4** | **76.0** |
| *Finetune multilingual model on all training sets (Translate-Train-All)* | | | | | | | | | | | | | | | | |
| XLM-R *reimpl.* | 85.4 | **81.5** | 82.0 | 80.7 | 80.2 | 81.2 | 78.9 | 78.4 | **77.6** | 79.9 | 77.6 | **79.5** | 75.8 | 73.4 | 72.3 | 79.0 |
| XLM-V | **85.6** | **81.5** | **82.1** | **81.5** | **80.7** | **81.5** | **79.6** | **78.7** | **77.6** | **80.0** | **77.7** | **79.5** | **77.0** | **74.3** | **73.9** | **79.4** |

Table 3: XLM-V outperforms XLM-R on cross-lingual transfer on every language in XNLI with outsized improvements on the lower-resource languages, Swahili and Urdu. We observe similar improvements on translate-train-all. The model is trained for 12 epochs (2 epochs for translate-train-all) on 8 A100 GPUs with float16 precision. We use a learning rate of 7.5e-6 with a max sequence length of 256, a batch size of 16, no weight decay, and no warmup.

| Model | aym | bzd | cni | gn | hch | nah | oto | quy | shp | tar | AVG |
|---|---|---|---|---|---|---|---|---|---|---|---|
| XLM-R *reimpl.* | 36.6 | 39.6 | 40.5 | 41.6 | 38.8 | 40.2 | 39.4 | 38.7 | 42.7 | 37.6 | 39.6 |
| XLM-V | **39.9** | **41.5** | **41.7** | **58.8** | **40.7** | **44.7** | **42.1** | **56.9** | **46.5** | **41.2** | **45.4** |
| Tok. Length (*rel.*) | -10.8% | -11.6% | -11.9% | -16.5% | -6.5% | -10.7% | -8.4% | -18.4% | -10.9% | -9.1% | -11.5% |

Table 4: We show the zero-shot cross-lingual transfer results on Americas NLI (trained on English and evaluated on the unseen languages). Our model, XLM-V, outperforms XLM-R by a wide margin with outsized improvements on Quechua and Guaraní. Tok. Length (*rel.*) refers to the relative difference in the average number of tokens (post-tokenization) between XLM-R and XLM-V. XLM-V consistently outputs shorter sequences post-tokenization. The model is trained for 12 epochs on 8 A100 GPUs with float16 precision. We use a learning rate of 7.5e-6 with a max sequence length of 256, batch size of 16, no weight decay, and no warmup.

TyDiQA, and XQuAD), named enitity recognition (WikiAnn), and low resource language tasks (AmericasNLI, MasakhaNER). All tasks leverage crosslingual transfer from English-only finetuning and are trained using float16 precision with the AdamW optimizer (Loshchilov and Hutter, 2017). We use hyperparameters selected based on the best English performance on the dev set,[3] and finally evaluate on the test set. We compile all of our results in Table 2 for XLM-V and XLM-R. We also include results for XLM (Lample and Conneau, 2019) for additional context.

Table 2 shows that XLM-V outperforms our re-implementation of XLM-R on all datasets by an average of 3.5 points absolute (we compute the average result using either the accuracy or F1 of each task). In Table 3, we show that XLM-V outperforms XLM-R on all languages in cross-lingual transfer (training on English and evaluating on other languages) with similar improvements on translate-train-all (finetuning the model on both the English and translated training sets). In particular, we find that XLM-V consistently outperforms XLM-R on low-resource languages. For example, in Table 3, we observe a 4.7% and 2.9% accu-

racy improvement on Swahili (sw) and Urdu (ur) on XNLI. Similarly, we show an average gain of 11.2% F1 on MasakhaNER, a low-resource African language NER dataset.

In Table 4 we show that XLM-V not only consistently outperforms XLM-R on Americas NLI in zero-shot crosslingual transfer but is able to obtain 18.2% absolute F1 improvement on Quechua (quy) and 17.2% absolute improvement on Guaraní (gn). Interestingly, Quechua and Guaraní are also the two languages with the largest relative drop in average token count per sentence – suggesting that these languages are over-tokenized by XLM-R.

## 6 Analysis

### 6.1 The Zipf ceiling

We explored training models with vocabulary sizes greater than 1M tokens but found that these models perform comparatively worse on downstream tasks. We visualize the diminishing utility of increasing the vocabulary size in Figure 3. Specifically, we create vocabularies with 500K, 1M, 1.5M, and 2M tokens using our methodology. Then, we use these vocabularies to tokenize the FLoRes-200 dataset. For vocabulary sizes of 500K, 1M, and 2M, we find that 99% of the content is covered by just 140,337, 197,817, and 243,832 unique tokens, respectively.

---

[3]For tasks trained on MNLI we follow Conneau et al. (2019) and select the checkpoint with the best average performance across all languages.

| Language | Tokenizer | Tokenized Output |
|---|---|---|
| zh | Original Sentence
XLM-R (250K)
XLM-R (1M)
Chung et al. (2020) (1M)
XLM-V (1M) | 剑桥大学本科生和研究生
['剑', '桥', '大学', '本科', '生', '和', '研究生']
['剑', '桥', '大学', '本', '科', '生', '和', '研究', '生']
['剑桥', '大学本科', '生', '和', '研究生']
['剑桥大学', '本科生', '和', '研究生'] |
| en, fr, es | Original Sentence
XLM-R (250K)
XLM-R (1M)
Chung et al. (2020) (1M)
XLM-V (1M) | narcolepsy narcolepsie narcolepsia
['na', 'r', 'cole', 'psy'] ['na', 'r', 'cole', 'psi', 'e'] ['na', 'r', 'cole', 'psi', 'a']
['na', 'rcole', 'psy'] ['na', 'rcole', 'psie'] ['na', 'rcole', 'psia']
['na', 'rcole', 'psy'] ['narco', 'lepsi', 'e'] ['na', 'rcole', 'psia']
['narco', 'le', 'psy'] ['narco', 'lepsi', 'e'] ['narco', 'lepsi', 'a'] |
| de | Original Sentence
XLM-R (250K)
XLM-R (1M)
Chung et al. (2020) (1M)
XLM-V (1M) | Betäubungsmittelverschreibungsverordnung
['Be', 'tä', 'ub', 'ungs', 'mittel', 'ver', 'schreibung', 's', 'ver', 'ordnung']
['Be', 'tä', 'ub', 'ungsmittel', 'ver', 'schreibung', 'ver', 'verordnung']
['Bet', 'äub', 'ungsmittel', 'ver', 'schreibung', 'sverordnung']
['Bet', 'äub', 'ungsmittel', 'ver', 'schreibung', 'sverordnung'] |

Table 5: We provide examples comparing tokenization using the XLM-V vocabulary against baselines. We find that our sentencepiece model reduces overtokenization and can be surprisingly good at splitting sentences into pseudo-meaningful segments out-of-the-box.

| Model | vi | zh | fr | de | en | xho | tel | AVG |
|---|---|---|---|---|---|---|---|---|
| XLM-R (250K) | 34.3 | 28.5 | 37.5 | 33.9 | 29.1 | 43.9 | 38.8 | 43.6 |
| XLM-R (1M) | 33.5 | 31.7 | 34.8 | 31 | 26.8 | 40.3 | 41.6 | 41.4 |
| Chung et al. (2020) (1M) | 32.7 | 24.4 | 32.9 | 29.1 | 27.8 | **29.2** | **25.7** | 37.7 |
| XLM-V (1M) | **32.4** | **23.4** | **32.2** | **28.3** | **25.5** | 37.4 | 33.2 | 38.6 |

Table 6: Average number of tokens after tokenization on the FLoRes-200 dataset for several high, medium, and low resource languages. AVG denotes the average tokenized lengths per sentence across all 200 languages in Flores-200.

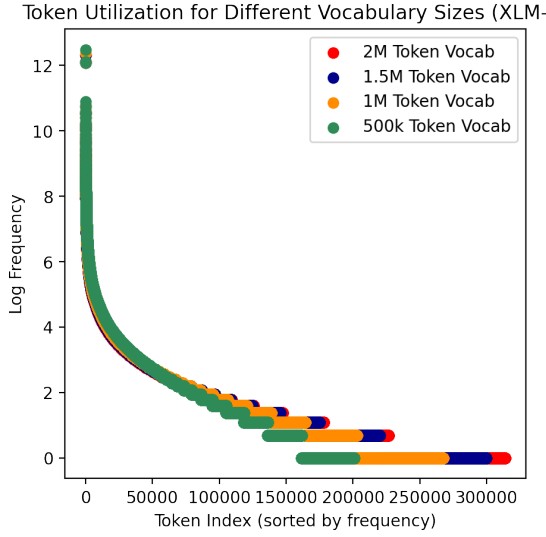

Figure 3: We compare the token utilization of each sentencepiece vocabulary on the FLoRes-200 dataset. We see diminishing returns as the size of the vocabulary is increased beyond 1M tokens.

We hypothesize that since the Unigram LM (Kudo and Richardson, 2018) algorithm used to construct the vocabulary iteratively prunes a large initial set, as discussed in Section 2, further expanding the vocabulary is equivalent to inheriting tokens from the long tail of a Zipfian distribution. These

token embeddings are problematic because they are trained on significantly less data during the course of MLM pretraining and will learn sub-optimal representations as a result. **As a consequence, vocabularies past a certain size will cease to improve model performance and can potentially degrade it**. A clear example of this is shown in Figure 2 where our model with a 1M token vocabulary outperforms its 1.5M token counterpart trained using an equivalent amount of data.

## 6.2 Qualitative improvements in tokenization

Table 5 shows a few tokenized examples from Chinese (zh), English (en), French (fr), Spanish (es), and German (de). For languages in the same cluster (en, fr, es), our method can separate shared roots (e.g. narco) from the same word in different languages. Notably, our method demonstrates a surprising ability to segment Chinese out-of-the-box, parsing out individual entities in the original phrase. For example, the XLM-V tokenizer is able to meaningfully break down the phrase 剑桥大学本科生和研究生, translated as *Cambridge University undergraduates and postgraduates*. Specifically, the output of the XLM-V tokenizer is 剑桥大学(Cambridge University), 本科生(undergraduates), 和(and), and 研究

生(postgraduates). **Qualitatively, our tokenizer frequently performs tokenizations that are semantically meaningful**, one possible contributor to the improved downstream performance.

### 6.3 Over-tokenization

Representing input data with fewer tokens can speed up inference, allow the model to make use of longer context, and help with over-tokenization for low-resource languages (Rust et al., 2020). Table 6 shows the average number of resulting tokens (post-tokenization) for several languages in FLoRes-200. On average, the XLM-V tokenizer returns fewer tokens for high and medium resource languages while Chung et al. (2020) returns the fewest tokens for low-resource languages. **Overall, XLM-V returns 11.5% fewer tokens compared to the baseline XLM-R tokenizer, meaning that input sequences are on average 11.5% shorter.**

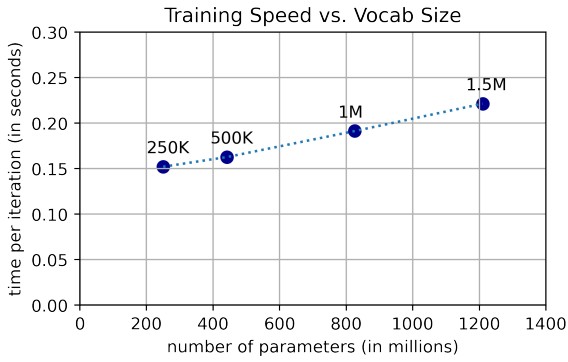

Figure 4: We track training speed vs. vocabulary size using a typical training setup on XNLI: one A100 GPU, a batch size of 16, sequence length of 128, and float16 precision. The text above each point denotes the vocabulary size.

### 6.4 Speed vs. size

For XLM-R, which has a vocabulary size of 250K tokens, the vocabulary embedding matrix contains 77% of the model's trainable parameters. For XLM-V, the 1M token vocabulary accounts for 93% of the model's trainable parameters. While scaling the vocabulary can markedly increase the number of trainable parameters in a model, we can treat it as an efficient form of conditional compute (Bengio et al., 2015): only a small fraction of the embedding matrix is used for any given input. We illustrate the relationship between the vocabulary size and training speed in Figure 4. **By increasing the vocabulary from 250K to 1M tokens, we can** **increase the number of trainable parameters by 3.3x with just a 25% increase in training time**.

## 7 Related work

### 7.1 Vocabulary-free models

In recent years, vocabulary-free models like ByT4 (Xue et al., 2022) and CANINE (Clark et al., 2022) have demonstrated on-par or better performance compared to their subword tokenization-based counterparts. However, one consistent drawback of these models is slower training and inference speed. For example, ByT5 is 6.4 to 9.5 times slower than mT5 (Xue et al., 2020) on classification tasks like XNLI. CANINE fares better, leveraging optimizations like lower input character dimensions and heavy down sampling, but still remains approximately 1.6 times slower than a comparable BERT baseline. On the other hand, simply using a larger sentencepiece vocabulary can improve downstream performance, increase the capacity of the model, and reduce the over-tokenization and coverage of low-resource languages all with a smaller impact on inference latency. We believe that both directions are useful areas of research and can be explored simultaneously.

### 7.2 Building larger vocabularies

Prior work on vocabulary expansion (Wang et al., 2019) sought to augment the vocabulary of existing models to address out-of-vocabulary (OOV) problems in multilingual settings. While these results are potentially useful in augmenting subword models like BERT, sentencepiece models by nature encounter significantly fewer OOVs.

More recent work on building larger vocabularies (Chung et al., 2020; Zheng et al., 2021) leverage tricks like lexical clustering and more principled methods for vocabulary allocation have tackled issues with over-tokenization and vocabulary coverage for low-resource languages. While compelling, these works are unfortunately limited by data (the models are trained on Wikipedia, a relatively small pretraining corpus) and scale (the largest vocabulary explored was 500K, only twice the size of the vocabulary in XLM-R). As such, the resulting models significantly under-perform the public XLM-R baseline. Our work seeks to combine and improve upon existing methods for building large-scale vocabularies, pretrain with substantially bigger datasets, and explore vocabularies of 1M tokens and beyond.

## 8 Conclusion

In this paper, we presented XLM-V, a multilingual language model with a 1M token vocabulary. We showed that our model outperforms XLM-R, has outsized gains on tasks in low-resource languages, results in semantically meaningful tokenizations, reduces average sequence length, and serves as an efficient form of conditional compute. In the future, we would like to further investigate the Zipf ceiling discussed in Section 6 by increasing the vocabulary beyond 2M tokens while also using more data. Another possible direction for future work is to explore larger multilingual vocabularies for autoregressive language models. Finally, further exploration with different clustering methods such as hierarchical clustering may prove both interesting and effective.

## Limitations

While the strengths of XLM-V are clear, there remains several scalability issues that are notable. First, while scaling the vocabulary is an efficient form of conditional compute, it can result in increased pre-training times due to the computational complexity of the softmax over the entire vocabulary. We believe these issues can be solved by adopting approximation techniques like adaptive softmax (Joulin et al., 2017) and adaptive inputs (Baevski and Auli, 2018). Additionally, scaling the vocabulary can also significantly increase the memory footprint of a model. However, we believe memory-related issues become less of a problem as we begin to work with larger models, where the number of non-embedding parameters vastly outweigh the size of the vocabulary embedding matrix.

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

# A    Appendix

# B    Appendix

We show the per-language results for each task we tested on. For the sake of reproducibility, we also provide the hyperparameters that we used to finetune the model for each task.

| Model | en | es | de | ar | hi | vi | zh | AVG |
|---|---|---|---|---|---|---|---|---|
| XLM-R *reimpl.* | 65.9 / 78.7 | 50.4 / 67.7 | 47.6 / 62.2 | 36.8 / 55.8 | 42.1 / 59.3 | 45.2 / 65.2 | 37.8 / 60.7 | 46.5 / 64.2 |
| XLM-V | 67.5 / 80.4 | 51.1 / 69.4 | 49.8 / 64.3 | 38.1 / 58.2 | 44.5 / 62.7 | 46.4 / 67.2 | 36.3 / 59.9 | 47.7 / 66.0 |

Table 7: MLQA results (EM/F1). The model is trained for 2 epochs on a single A100 GPU with float16 precision. We use a learning rate of 3e-5 with a max sequence length of 512, batch size of 6, no weight decay, and no warmup.

| Model | en | ar | bn | fi | id | ko | ru | sw | te | AVG |
|---|---|---|---|---|---|---|---|---|---|---|
| XLM-R *reimpl.* | 55.5/68.6 | 42.0/63.9 | 18.6/37.6 | 42.8/61.6 | 54.7/73.1 | 23.6/39.9 | 31.5/59.9 | 30.7/54.1 | 27.5/44.3 | 36.3/55.9 |
| XLM-V | 52.3/66.9 | 45.4/65.5 | 27.4/42.7 | 46.0/63.6 | 56.1/72.3 | 22.8/37.4 | 31.5/59.3 | 43.1/61.4 | 32.4/43.2 | 39.7/56.9 |

Table 8: TyDiQA-GoldP results (EM/F1). The model is trained for 8 epochs on a single A100 GPU with float16 precision. We use a learning rate of 3e-5 with a max sequence length of 512, batch size of 6, no weight decay, and no warmup.

| Model | en | es | de | el | ru | tr |
|---|---|---|---|---|---|---|
| XLM-R *reimpl.* | 72.1 / 83.5 | 58.5 / 76.5 | 57.6 / 73.0 | 55.4 / 72.2 | 56.6 / 73.1 | 52.2 / 68.3 |
| XLM-V | 72.9 / 84.2 | 60.3 / 78.1 | 57.3 / 75.1 | 53.5 / 72.4 | 56.0 / 73.2 | 51.8 / 67.5 |

| | ar | vi | th | zh | hi | AVG |
|---|---|---|---|---|---|---|
| XLM-R *reimpl.* | 49.2 / 65.9 | 53.5 / 72.9 | 55.7 / 66.3 | 55.5 / 65.3 | 49.8 / 57.7 | 56.0 / 71.3 |
| XLM-V | 51.2 / 67.5 | 53.7 / 73.1 | 56.9 / 67.0 | 53.5 / 63.1 | 51.9 / 69.4 | 56.3 / 71.9 |

Table 9: XQuAD Results (EM/F1). The model is trained for 2 epochs on a single A100 GPU with float16 precision. We use a learning rate of 3e-5 with a max sequence length of 512, batch size of 6, no weight decay, and no warmup.

| Model | ro | gu | pa | lt | az | uk | pl | qu | hu | fi | et | tr | kk | zh | my | yo | sw |
|---|---|---|---|---|---|---|---|---|---|---|---|---|---|---|---|---|---|
| XLM-R *reimpl.* | 73.5 | 62.9 | 53.6 | 72.7 | 61.0 | 72.4 | 77.5 | 60.4 | 75.8 | 74.4 | 71.2 | 75.4 | 42.2 | 25.3 | 48.9 | 33.6 | 66.3 |
| XLM-V | 73.8 | 66.4 | 48.7 | 75.6 | 66.7 | 65.7 | 79.5 | 70.0 | 79.5 | 78.7 | 75.0 | 77.3 | 50.4 | 30.2 | 61.5 | 54.2 | 72.4 |

| | th | ko | ka | ja | ru | bg | es | pt | it | fr | fa | ur | mr | hi | bn | el | de |
|---|---|---|---|---|---|---|---|---|---|---|---|---|---|---|---|---|---|
| XLM-R *reimpl.* | 5.2 | 49.4 | 65.4 | 21.0 | 63.1 | 76.1 | 70.2 | 77.0 | 76.9 | 76.5 | 44.6 | 51.4 | 61.5 | 67.2 | 69.0 | 73.8 | 74.4 |
| XLM-V | 3.3 | 53.0 | 69.5 | 22.4 | 68.1 | 79.8 | 74.5 | 80.5 | 78.7 | 77.6 | 50.6 | 48.9 | 59.8 | 67.3 | 72.6 | 76.7 | 76.8 |

| | en | nl | af | te | ta | ml | eu | tl | ms | jv | id | vi | he | ar | AVG |
|---|---|---|---|---|---|---|---|---|---|---|---|---|---|---|---|
| XLM-R *reimpl.* | 83.0 | 80.0 | 75.83 | 49.2 | 56.3 | 61.9 | 57.2 | 69.8 | 68.3 | 59.4 | 48.6 | 67.7 | 53.2 | 43.8 | 61.3 |
| XLM-V | 83.4 | 81.4 | 78.3 | 51.8 | 54.9 | 63.1 | 67.1 | 75.6 | 70.0 | 67.5 | 52.6 | 67.1 | 60.1 | 45.8 | 64.7 |

Table 10: NER Results. The model is trained for 10 epochs on a single A100 GPU with float16 precision. We use a learning rate of 2e-5 with a max sequence length of 128, batch size of 32, no weight decay, and no warmup.

| Model | amh | hau | ibo | kin | lug | luo | pcm | swa | wol | yor | AVG |
|---|---|---|---|---|---|---|---|---|---|---|---|
| XLM-R *reimpl.* | 25.1 | 43.5 | 11.6 | 9.4 | 9.5 | 8.4 | 36.8 | 48.9 | 5.3 | 10.0 | 20.9 |
| XLM-V | 20.6 | 35.9 | 45.9 | 25.0 | 48.7 | 10.4 | 38.2 | 44.0 | 16.7 | 35.8 | 32.1 |

Table 11: We show the zero-shot cross-lingual transfer results on MasakhaNER (trained on English and evaluated on the unseen languages). The model is trained for 10 epochs on a single A100 GPU with float16 precision. We use a learning rate of 2e-5 with a max sequence length of 128, batch size of 32, no weight decay, and no warmup.