# OpenReview forum: "XLM-V: Overcoming the Vocabulary Bottleneck in Multilingual Masked Language Models"
_EMNLP/2023/Conference — EMNLP 2023 Main_

### Official Review · Reviewer_hL9S · 2023-07-31

**Soundness:** 4

**Excitement:**

3: Ambivalent: It has merits (e.g., it reports state-of-the-art results, the idea is nice), but there are key weaknesses (e.g., it describes incremental work), and it can significantly benefit from another round of revision. However, I won't object to accepting it if my co-reviewers champion it.

**Paper Topic And Main Contributions:**

This paper focus on multilingual large language models. Specifically, they improve the previous released masked language model XLM-R by using a much larger vocabulary that has 1 million tokens. They also did various experiments on various multilingual benchmarks to show that the proposed model is much better than previous XLM-R.

Contribution:

(1): propose a new method to construct the large (one million) vocabulary with lexical clustering.

(2): based on this newly constructed vocabulary, they followed previous XLM and trained a new model called XLM-V.

(3): experimental results on various benchmarks including XNLI, FloRes-200, MLQA etc demonstrate the superiority of their new model.




**Questions For The Authors:**

Thanks for this great work, I have the following questions for you:

1: except the lexical clustering, do you tried some more naive method such as directly compute the vocabulary overall all the corpus?

2: to cluster the language, except the k-means, have we tried any other clusters algorithm? or have we changed the cluster number to other number?

3: do you have any insights why the approximated softmax doesn't work too much or unstable for pre-training? I think if we want to move to 2 million tokens, then we have to solve the softmax problem first.


**Reasons To Accept:**

This paper explore an interesting idea: scaling the vocabulary in LLM to one million. And experimental results demonstrate the superiority of their new model.

**Reasons To Reject:**

I don't find any obvious reasons to reject this paper.

**Reproducibility:**

3: Could reproduce the results with some difficulty. The settings of parameters are underspecified or subjectively determined; the training/evaluation data are not widely available.

**Reviewer Confidence:**

4: Quite sure. I tried to check the important points carefully. It's unlikely, though conceivable, that I missed something that should affect my ratings.

---

### Official Review · Reviewer_rngv · 2023-08-05

**Soundness:** 4

**Excitement:**

4: Strong: This paper deepens the understanding of some phenomenon or lowers the barriers to an existing research direction.

**Paper Topic And Main Contributions:**

The authors propose XLM-V, which is a multilingual language model with a larger vocabulary size compared to the XLM-R model. Their approach scales up the size of the vocabulary by reducing the emphasis on sharing tokens between languages that have limited word lexical overlap and allocating vocabulary space to ensure sufficient coverage for each individual language.

**Reasons To Accept:**

The idea is good, and the method is straightforward. The paper is easy to follow and well-written. The authors tackle the vocabulary bottleneck and handle the over-tokenization for low-resource languages.

**Reasons To Reject:**

I do not have a reason to reject it. But I am curious about the performance of the XLM-V model on the GLUE benchmark dataset. The authors evaluate XLM-V on 3 types of downstream tasks (e.g., NLI, NER and QA) with different datasets, however, as XLM-R is the model that they direct compare with, I wonder if XLM-V can also beat XLM-R on GLUE benchmark datasets.

**Reproducibility:**

4: Could mostly reproduce the results, but there may be some variation because of sample variance or minor variations in their interpretation of the protocol or method.

**Reviewer Confidence:**

3: Pretty sure, but there's a chance I missed something. Although I have a good feel for this area in general, I did not carefully check the paper's details, e.g., the math, experimental design, or novelty.

---

### Official Review · Reviewer_BY56 · 2023-08-08

**Soundness:** 4

**Excitement:**

4: Strong: This paper deepens the understanding of some phenomenon or lowers the barriers to an existing research direction.

**Paper Topic And Main Contributions:**

This paper describes XLM-V, a new entry in the field of multilingual MLMs, with innovations in both the size of the shared vocabulary and the ways in which that vocabulary is shared (or, importantly, sometimes not shared). The key innovations are:

1 -- an expansion of the model's shared subword vocabulary to roughly 1M tokens.
2 -- incorporation of a lexical clustering approach (following Chung et al. 2020) which is used to first identify clusters of languages with a higher-than-average degree of lexical overlap, and then to encourage vocab sharing between clustered languages, and discourage vocab sharing between non-clustered languages [roughly put]
3 -- incorporating of a vocabulary allocation measure (from Zheng et al. 2021) in order to decide how much of the overall vocabulary to devote to each language cluster

For a range of tasks, on a range of languages, the model outperforms both smaller vocabulary models and reimplementations of earlier models at a similar vocabulary size, but either without the clustering and allocation steps described above, or with only one or the other.

Increasing the vocabulary size of course increases both the memory and compute demands of the model, but the authors find that compute time does not in fact scale at the same rate of the increase vocabulary size, largely because the development of specific shared vocabularies for lexically-similar languages means that most of the vocabulary is unused for any given input sample.

This work presents a logical next step in multilingual language modeling and should be useful for many researchers, particularly those working on low-resource languages, as this approach promises to combat some of the problems associated with poor vocabulary representation for diverse languages.

**Questions For The Authors:**

A. lines 217-220: First, a clarification. I assume that the set of languages here is the same as for Chung et al., namely those from CC100 -- is this accurate? It might be worth pointing out that the ideal number of clusters should be expected to vary not based on the number of languages in the set of 100+ languages, but rather on the identity of those languages and their respective similarities with one another. How do you expect this clustering approach might scale as more languages get included? And more generally, how do you expect the model might scale as you reach beyond 100 languages?

B. Overall, it would be nice to have a table showing not only the number of languages covered by each data set, but also which languages are in each data set, and where they do and do not overlap.

C. Regarding bold-facing of results in all results tables - there are some inconsistencies. For example, in Table 2, the bold-faced value for XNLI is not in fact the highest score. In some cases, matching scores are both highlighted, in some only the XLM-V row is highlighted. More generally, has any significance testing been done?

D. Figure 3 is hard to interpret visually - it made sense to me only after reading the textual description.

E. The table for MNER in the appendix shows a different picture from what the text suggests -- there are quite a few languages for which XLM-V is significantly outperformed by XLM-R. It would be nice to see this addressed, analyzed, and discussed in the body of the paper.

F. line 451: do you actually mean to say that 'sverordnung' is the correct German word? In case this is new information, the noun is 'Verordnung', and the 's' is commonly used for linking words within German compound nouns. I'm assuming that this is not new information, though, in which case I think the text around this example should be softened. Minor quibble.



**Reasons To Accept:**

This is a strong paper presenting sensible developments for multilingual MLMs, and with improved performance for many tasks and languages.

1. The approach addresses an ongoing problem associated with multilingual models and has some promise for combatting the vocabulary bottleneck issue.

2. The experimental work is carefully and thoroughly carried out and reported. In addition, the paper includes a reasonable amount of analysis (we always like more!).

3. The paper is well-written and well-organized.

4. The results on low-resource data sets are especially exciting and encouraging!

**Reasons To Reject:**

Mostly I only have small quibbles with this paper (see comments and questions below) and would be happy to see it accepted. However, there's one claim I find unconvincing (or at least not especially convincingly demonstrated). This is the claim that the tokenizations produced by XLM-V are qualitatively better, linguistically more sensible, and/or semantically more meaningful than those produced by earlier models.

The only demonstration given for this claim comes from the three examples presented in Table 5. Of these examples, I see mixed results (with the caveat that I'm not able to assess the Chinese tokenization). While I would agree that the segmentations produced for variant forms of the word 'narcolepsy' look better for XLM-V than for other models, I'm not sure the same is true for the German example. For the German example, XLM-V does better toward the end of the compound than earlier models (arguably -- why segment off 'ver' for 'verschreibung' but not for 'verordnung'?), its segmentation for the beginning of the compound isn't much better (e.g. 'ungsmittel' makes little sense).

More generally, though, I'm not quite willing to accept this claim based on a few selected examples. The paper could really be improved with a more extensive analysis, perhaps comparing to outputs of morphological analyzers for some languages, and to word segmentation systems for others (such as Chinese)?

All this said, this alone is not a rejection-worthy shortcoming.

**Reproducibility:**

4: Could mostly reproduce the results, but there may be some variation because of sample variance or minor variations in their interpretation of the protocol or method.

**Reviewer Confidence:**

4: Quite sure. I tried to check the important points carefully. It's unlikely, though conceivable, that I missed something that should affect my ratings.

**Typos Grammar Style And Presentation Improvements:**

* lines 306-314: please add # of languages in MLQA

* Table 2: please add description of what 'EM' stands for

* lines 329-335: please add # of languages for NER

* lines 336-340: it would be nice to have a description of ANLI languages similar to that provided for MNER

* References: please fix capitalization issues!

---

### Meta-Review · Area_Chair_QLue · 2023-09-19

**Recommendation:** 5

**Metareview:**

This work studies the impact of the vocabulary size on a XLM  model, together with a lexical cluster approach and encouraging vocabulary share between clustered languages. This leads to overall performance boosts on a large variety of benchmarks. It is also remarkable that the authors did the experiments on truly low resource languages. The approach was exciting for all reviewers and no major objections to include this paper in to the main conference were made.

---

### Decision · Program_Chairs · 2023-10-07

**Decision:**

Accept-Main

**Comment:**

This work studies the impact of the vocabulary size on a XLM  model, together with a lexical cluster approach and encouraging vocabulary share between clustered languages. This leads to overall performance boosts on a large variety of benchmarks. It is also remarkable that the authors did the experiments on truly low resource languages. The approach was exciting for all reviewers and no major objections to include this paper in to the main conference were made.